# Accounting for Healthcare-Seeking Behaviours and Testing Practices in Real-Time Influenza Forecasts

**DOI:** 10.3390/tropicalmed4010012

**Published:** 2019-01-11

**Authors:** Robert Moss, Alexander E. Zarebski, Sandra J. Carlson, James M. McCaw

**Affiliations:** 1Centre for Epidemiology and Biostatistics, Melbourne School of Population and Global Health, The University of Melbourne, Parkville 3052, Australia; jamesm@unimelb.edu.au; 2Department of Zoology, The University of Oxford, Oxford OX1 3SZ, UK; aezarebski@gmail.com; 3Hunter New England Population Health, Wallsend 2287, Australia; Sandra.Carlson@hnehealth.nsw.gov.au; 4School of Mathematics and Statistics, The University of Melbourne, Parkville 3052, Australia; 5Murdoch Children’s Research Institute, The Royal Children’s Hospital, Parkville 3052, Australia; 6Victorian Infectious Diseases Reference Laboratory Epidemiology Unit, Peter Doherty Institute for Infection and Immunity, The Royal Melbourne Hospital and The University of Melbourne, Melbourne 3000, Australia

**Keywords:** influenza, epidemics, forecasting, public health, surveillance, ascertainment

## Abstract

For diseases such as influenza, where the majority of infected persons experience mild (if any) symptoms, surveillance systems are sensitive to changes in healthcare-seeking and clinical decision-making behaviours. This presents a challenge when trying to interpret surveillance data in near-real-time (e.g., to provide public health decision-support). Australia experienced a particularly large and severe influenza season in 2017, perhaps in part due to: (a) mild cases being more likely to seek healthcare; and (b) clinicians being more likely to collect specimens for reverse transcription polymerase chain reaction (RT-PCR) influenza tests. In this study, we used weekly Flutracking surveillance data to estimate the probability that a person with influenza-like illness (ILI) would seek healthcare and have a specimen collected. We then used this estimated probability to calibrate near-real-time seasonal influenza forecasts at each week of the 2017 season, to see whether predictive skill could be improved. While the number of self-reported influenza tests in the weekly surveys are typically very low, we were able to detect a substantial change in healthcare seeking behaviour and clinician testing behaviour prior to the high epidemic peak. Adjusting for these changes in behaviour in the forecasting framework improved predictive skill. Our analysis demonstrates a unique value of community-level surveillance systems, such as Flutracking, when interpreting traditional surveillance data. These methods are also applicable beyond the Australian context, as similar community-level surveillance systems operate in other countries.

## 1. Introduction

Public health surveillance systems provide valuable insights into disease incidence, which can inform preparedness and response activities [1]. The data obtained from these systems must be interpreted appropriately, which requires an understanding of the characteristics of each system. For example, there is a well-established need to “understand the processes that determine how persons are identified by surveillance systems in order to appropriately adjust for the biases that may be present” [1]. The importance of understanding how people are identified by a surveillance system is particularly problematic in the context of near-real-time influenza forecasting [2,3,4,5,6,7,8,9,10], because healthcare-seeking behaviours and clinical decision-making are dynamic [11,12] and are subject to acute influences (e.g., media coverage [13]). The challenge that this poses is further compounded by delays in data collection and reporting, which reduce forecast performance [9,14,15]. In the event of a pandemic, changes in patient and clinician behaviours are likely to be even more pronounced, with ramifications not only for interpreting surveillance data [3], but also for delivering effective and proportionate public health responses [16]. For influenza and influenza-like illnesses (ILI), the Flutracking online community surveillance system offers unique perspectives on disease incidence and severity in Australia, as well as on healthcare-seeking behaviours and clinical decision-making [17,18,19,20].

The 2017 influenza season was particularly severe in most of Australia, with an unprecedented number of laboratory-confirmed influenza cases, and high consultation, hospitalisation, and mortality rates [21]. The 2017 season presented a very real challenge to public health staff in Australia for a number of reasons, among which were the unprecedented scale of the surveillance data and the delays in reporting that this entailed. In the subsequent 2017/2018 northern hemisphere influenza season, the dominant strain was frequently referred to as the “Australian flu” [22].

We have previously adapted epidemic forecasting methods to the Australian context [23]. Notable features of this context include: more than half of the population reside in the five largest cities (Sydney, Melbourne, Brisbane, Perth, and Adelaide), which are sparsely distributed across a landmass approximately the same size as continental USA; ILI data are available from sentinel surveillance systems and public health services, but on a much smaller scale than, e.g., the ILINet system in the USA; and laboratory-confirmed influenza infections data are available, but testing denominators are typically only provided by public laboratories (which comprise only a small proportion of all laboratory testing). In previous work, we applied these forecasting methods to individual Australian cities, and used them to compare multiple surveillance systems [3], developed model-selection methods to measure the benefit of incorporating climatic effects [24], and deployed these forecasts in real-time in collaboration with public health staff [9,10]. Most recently, we used these methods to quantify the impact of delays in reporting and to highlight challenges in accounting for population and clinician behaviour changes in response to a severe season [10].

In 2017, we produced weekly near-real-time influenza forecasts for the capital cities of three Australian states: Sydney (New South Wales), Brisbane (Queensland), and Melbourne (Victoria). As described previously, we performed a one-off re-calibration of our real-time influenza forecasts (2–5 weeks prior to the observed peaks), which improved the forecast performance [10]. This was a manual process and was informed by qualitative expert opinion rather than by quantitative methods. In the 2018 influenza season, we also began producing weekly near-real-time influenza forecasts for Perth, the capital city of Western Australia.

Here, we investigated how the use of community-level ILI surveillance data, as captured by the Flutracking surveillance system [17,18,19,20], may allow us to re-calibrate our forecasts in real-time based on quantitative measures, and potentially improve forecast performance. We used the 2014–2016 seasons in these four cities for calibration, and the 2017 season in these four cities as a case study.

## 2. Results

Retrospective analysis revealed that in the 2017 influenza season there was a marked increase in the probability that Flutracking participants with ILI symptoms (defined in Section 4.1) would seek healthcare and have a specimen collected for testing, relative to previous influenza seasons, in all states except Western Australia (see Table 1 and Figure 1). In the other influenza seasons (2014–2016), there was no such increase; this probability remained very similar to the average of the 2014 season (horizontal dashed lines in Figure 1). Queensland was the one exception, where this probability did increase in 2015 and 2016, but not by the same scale as observed in 2017.

Consistent with the Flutracking analysis shown in Figure 1, the 2017 influenza season was substantially larger than previous influenza seasons in all cities except Perth (Figure 2). In contrast to the other cities, the data for Perth comprise only a subset of all influenza case notifications for that city and, relative to the other cities, contain a higher proportion of hospitalised cases. These data are therefore likely to be less sensitive to behaviour changes (see Section 4.3 for details).

### 2.1. Relating Flutracking Trends to the General Population in 2014–2016

We hypothesised that these trends in influenza testing would also be reflected in the general population, but that there may be a difference in scale between the behaviour changes reported in the Flutracking surveys and those of the general population. We introduced a scaling factor κ to characterise the relationship between the testing probability obtained from the Flutracking surveys and the testing probability of the general population (Figure 3). When κ=1, the change in the population testing probability $p_{\mathrm{id}}\left(w\right)$—relative to its initial value $p_{\mathrm{id}}\left(0\right)$—is the same as the change in the Flutracking testing probability $P\left(\mathrm{Test}|\mathrm{ILI},\mathrm{week}=w\right)$ relative to its initial value $\mu_{0}=P\left(\mathrm{Test}|\mathrm{ILI},\mathrm{week}=0\right)$. When κ>1, the population testing probability is more sensitive to changes in the Flutracking testing probability. When κ=0, we recover the *non-calibrated forecasts*—i.e., the forecasts obtained without accounting for any behaviour changes. We used retrospective forecasts to identify an optimal value for this scaling parameter.

We produced retrospective weekly influenza forecasts for the 2014, 2015, and 2016 influenza seasons for the capital cities of four Australian states: Sydney (New South Wales), Brisbane (Queensland), Melbourne (Victoria), and Perth (Western Australia). At each week, the forecasts were calibrated based on the Flutracking survey data, by adjusting the *testing probability*
$p_{\mathrm{id}}\left(w\right)$ at each week *w*. This testing probability was used to relate disease incidence in epidemic simulations to reported surveillance data (see Section 4.4). We evaluated how this calibration affected forecast performance, according to four different forecasting targets:“Now-casts”: How well forecasts predicted the observation for the forecasting date itself.“Next 2 weeks”: How well forecasts predicted observations 1–2 weeks after the forecasting date.“Next 4 weeks”: How well forecasts predicted observations 1–4 weeks after the forecasting date.“Next 6 weeks”: How well forecasts predicted observations 1–6 weeks after the forecasting date.

Forecast performance was evaluated using Bayes factors, which are the ratio of the likelihood that the available data support two competing hypotheses [24,25]. In this study, we used the ratio of the calibrated forecast likelihood (where κ=k) to the non-calibrated forecast likelihood, averaged over all of the forecasting weeks, and reported the logarithm $\tilde{b_{k}}$ of this average, which we denote the “mean performance” (see Section 4.5 for details). Where $\tilde{b_{k}}>1$ the *calibrated forecasts* are considered to be more strongly supported by the data, and where $\tilde{b_{k}}<-1$ the *non-calibrated forecasts* are considered to be more strongly supported. To measure forecast performance over multiple years, we normalised the mean performance $\tilde{b_{k}}$ in each year and reported the mean $\hat{b_{k}}$ of these annual values, which we denote the “mean normalised performance”.

There were consistent trends across these four targets in each city, and substantial differences in performance between the cities, as summarised in Figure 4. For values of κ between 1 and 5, the calibrated forecasts strongly out-performed the non-calibrated forecasts in both Brisbane and Melbourne. Because the 2017 season was much larger (relative to previous seasons) in Melbourne than in Brisbane, the non-calibrated forecasts performed worse in Melbourne than they did in Brisbane. Accordingly, greater performance improvements were possible in Melbourne. Over a similar range, the calibrated forecasts showed no improvement in Sydney, and performed much worse in Perth than the non-calibrated forecasts. Note that, in Sydney, for 0<κ≤2 the forecast performance was not meaningfully better *or* worse than the non-calibrated forecasts (κ=0). For larger values of κ, the calibrated forecasts were out-performed by the non-calibrated forecasts in all cities and for all targets. Based on the mean *normalised* performance in all cities *except* Perth, we identified an optimal value κ=1.75 (Figure 5). Note that this value is greater than 1, which suggests that the behaviour of the general population exhibits larger changes than that of the Flutracking cohort.

### 2.2. Incorporating Flutracking Trends into 2017 Forecasts

The next question was whether the optimal value of κ, as chosen based on the results for 2014–2016, would have led to improved forecast performance in 2017. In other words, are these findings of value in a real-time context?

In a retrospective analysis, we measured the forecast performance for the 2017 influenza season over the same range of values for κ, in order to evaluate how well the chosen value of κ=1.75 would have performed. Note that this represents what would have been possible in real-time in the 2017 season. The values of the calibrated testing probabilities $p_{\mathrm{id}}\left(w\right)$ are shown in Figure 6, and the resulting performance according to each of the forecasting targets is shown in Figure 7.

When we set κ=1.75 (the optimal value according to the 2014–2016 forecasts) forecast performance in 2017 was near-optimal according to all forecasting targets in all cities except Perth (Figure 7). This was even true of Sydney, where similar values of κ in the 2014–2016 seasons had minimal impact on forecast performance. Note that the relative magnitudes of the performance according to each target is to be expected, since each target considers a different number of observations, and so a ten-fold improvement in predicting a single observation causes a hundred-fold improvement in predicting two observations.

For illustration, a subset of the Brisbane 2017 forecasts are shown in Figure 8, which shows how well the forecast predictions match the reported data over a range of values for κ, and at different times during the 2017 season.

## 3. Discussion

### 3.1. Principal Findings

Incorporating the behavioural trends estimated from the Flutracking self-reported influenza tests into the 2014–2016 forecasts greatly improved influenza forecast performance in both Brisbane and Melbourne, but had negligible effect on performance in Sydney and substantially reduced performance in Perth (this was not unexpected, see the next section). We explored the relationship between the Flutracking survey data and the testing probability in the general population, identifying an optimal scaling value of κ=1.75. This should be interpreted as meaning that behaviour changes in the general population were 75% larger than the behaviour changes characterised in the Flutracking survey data (where “behaviour changes” refers to healthcare-seeking behaviours and to clinician testing practices).

Using this same value for the scaling parameter in the 2017 forecasts greatly improved performance in both Brisbane and Melbourne, and also in Sydney. When we explored a wide range of values for the scaling parameter, these performance improvements were found to be near-optimal. Performance in Perth was reduced, but not as substantially as in the 2014–2016 forecasts.

Similar performance improvements would have been obtained in the 2014–2016 seasons, and in the 2017 season, if we had instead assumed that there was no difference between Flutracking participants and the general population (i.e., had we set κ=1). This provides evidence that, with the possible exception of Perth, the behaviour changes characterised by the Flutracking data are not strongly biased.

These findings demonstrate that the insights into healthcare behaviours that are provided by community-level surveillance systems such as Flutracking can greatly improve the performance of influenza forecasting systems. This is particularly true in the event of an unusual epidemic, such as those observed in the 2017 Australian influenza season, where these behaviours may change substantially relative to previous seasons, and also over the course of the epidemic. Improvements in forecast performance are also likely in the event of a pandemic, where the perception of risk is likely to differ markedly from seasonal epidemics [26,27]. Incorporating these kinds of behavioural insights may also improve the performance of other infectious disease forecasting systems.

### 3.2. Study Strengths and Weaknesses

It is intrinsically difficult to incorporate the dynamics of human behaviour into an infectious disease modelling framework, as we have attempted here, because it involves many complexities and sources of uncertainty [11,28]. Retrospective analyses of data from multiple surveillance systems have previously been used to estimate disease severity in, e.g., the 2009 H1N1 pandemic in England [29], and evidence synthesis in a modelling framework is an active area of research [30]. However, the use of one surveillance system to calibrate the interpretation of another in real-time is, to our knowledge, a novel and significant contribution of this study.

The methods outlined here are also applicable beyond the Australian context. Surveillance systems similar to Flutracking operate around the world, including Influenzanet in many European countries [31] and Flu Near You in North America [32].

We have used data from multiple cities that routinely experience seasonal influenza epidemics, but which are geographically and climatically diverse. As the results for Perth show, incorporating behaviour changes into the forecasts is not guaranteed to improve forecast performance. However, it does appear that the impact is reasonably consistent within a specific context, such as the same city from one year to the next. This was not an issue of sample size, since there are a similar number of participants from Western Australia as there are for other Australian states (see Table 1). Instead, it is likely due to differences in the surveillance data (detailed in Section 4.3). The Perth data were obtained from the public pathology laboratory (PathWest) and did not include serologic test results or data from private laboratories. Furthermore, these data contained a much higher proportion of hospitalised cases (40–60%) than did the data for the other cities in this study. Since people with severe disease will most likely seek healthcare, regardless of behavioural influences, we would expect these data to be less sensitive to behaviour changes in the general community.

It may also reflect, in part, that Western Australia typically experiences an influenza season that is markedly different from the rest of the country [21]. The testing probability for Western Australia was near-constant and differed little across the 2014–2017 seasons (Figure 1), and thus it is possible that the “signal” in the Flutracking data for Western Australia was, in fact, noise. In contrast, for Sydney there was no clear signal in the Flutracking data for the 2014–2016 seasons (Figure 1) and no improvement in forecast performance in these years (Figure 4). However, there was a marked change in the testing probability for Sydney in the 2017 season (Figure 1) and accounting for this yielded a substantial increase in forecast performance (Figure 7).

### 3.3. Meaning and Implications

Infectious diseases that primarily cause mild (if any) symptoms are difficult to observe, especially in a *consistent manner*. The visibility of such cases to a surveillance system is sensitive to human behaviours that are primarily based on the perception of risk, which is subject to many influencing factors and can change rapidly [26,27]. Seasonal influenza is an archetypal example, which routinely presents a substantial challenge to healthcare systems in temperate climates, and whose clinical severity profile can differ substantially from one year to the next, and from setting to setting (e.g., city to city). Infectious disease forecasting is now beginning to establish itself as a useful decision-support tool for public health preparedness and response activities, particularly for seasonal influenza [2,3,4,5,6,7,8,9,10,14,15]. One of the many challenges for this field is to account for how behaviour changes affect surveillance data, and how these data should be interpreted [7,8,9,10]. Many sources of “non-traditional” data have been explored for their potential to improve infectious disease forecast performance, such as internet search engine queries and social media trends [33,34]. While these non-traditional data may indeed provide useful and valid insights, it is unclear how representative they are of disease burden [12,35], especially since those populations that experience the greatest burden of disease are also likely to be the most under-represented in these data [36]. Here, we have used a community-level surveillance system that is explicitly designed to capture influenza-like illness activity outside of healthcare systems, and associated healthcare-seeking behaviour. We argue that the Flutracking survey data provide a much more direct and meaningful quantification of how visible cases are to routine surveillance systems, and have shown here that incorporation of these data yield substantial improvements in forecast performance in what was an unprecedented influenza season.

During the early months of the 2017 Australian influenza season (up to the end of July), at which time we were not using Flutracking data to calibrate our forecasts [10], our near-real-time influenza forecasts were predicting the epidemic peaks to be similar in size to previous years, and to occur in early August. These would have been relatively early peaks—although not unusually so—and these predictions were *consistent with expert opinion at that time*. Had we been in a position to incorporate the Flutracking survey data into our forecasts at that time using the methods introduced in this study, we would have had advance warning that, contrary to our original predictions *and* expert opinion, we should have expected later (and, therefore, also larger) epidemic peaks. An early warning of this kind, which indicates that the current situation appears to be unusual and is inconsistent with “usual” expectations, is in itself a valuable decision-support tool, even in the absence of an accurate prediction of what will eventuate.

Seasonal influenza epidemics, and the threat of pandemic influenza, continue to present substantial challenges for public health preparedness and response in temperate climates, some 100 years after the 1918–1919 influenza pandemic. Further complicating these activities are human behaviour changes, which confound the interpretation of surveillance data. However, as we have shown here, community-level surveillance is able to provide key insights into behaviour changes that can substantially improve near-real-time epidemic prediction and assessment. These advantages are of very real value to public health preparedness and response activities in the 21st century.

## 4. Materials and Methods

Access to the influenza surveillance data and the Flutracking survey data was approved by the Health Sciences Human Ethics Sub-Committee, Office for Research Ethics and Integrity, The University of Melbourne (Application ID 1646516.3).

### 4.1. Flutracking Survey Data

Flutracking weekly surveys were completed online voluntarily by community members in Australia and New Zealand. Participants responded to the weekly survey from April to October each year, and submitted surveys for themselves, and/or on behalf of other household members. The cohort of participants was maintained from year to year, unless participants unsubscribed. Participants were recruited online through workplace campaigns, current participants, the Flutracking website, traditional media and social media promotion.

Weekly counts of completed Flutracking surveys, self-reported ILI, and self-reported influenza tests were obtained from 2014 to 2017 for participants who had a usual postcode of residence in New South Wales, Victoria, Queensland, and Western Australia (see Table 1 for descriptive statistics of Flutracking data).

A participant with ILI was defined as having both self-reported fever and cough in the past week (ending Sunday). Any responses of “don’t know” to “fever”, “cough”, or “time off work or normal duties” variables were removed from Flutracking data prior to data provision. This removed 0.9% of all surveys. Only new ILI symptoms were counted in the analysis. New symptoms were defined as the first week of a participant reporting ILI symptoms (where there were consecutive weeks of reporting ILI symptoms). If a participant reported ILI symptoms in one week, and then reported at least one week of no symptoms, followed by another report of symptoms, then this second symptom report was considered a new incident of ILI.

Participants with ILI who reported seeking health advice for their symptoms were asked whether they had been tested for influenza. On average, fewer than half of participants with ILI reported that they sought health advice. Any self-reported influenza tests during an ILI incident were recorded in the same week of the new ILI symptoms (even if the test result was reported during the second, third or fourth week of illness). These self-reported tests are likely influenced by recall bias and underestimate the true numbers; participants are more likely to recall having a specimen collected if they are subsequently notified of the test result, and they are more likely to be notified if the test result is positive. We also acknowledge that Flutracking participant health seeking and testing behaviours may not be representative of the general Australian population. Flutracking has disproportionately high numbers of participants who are female, of working age, health care workers, have higher educational achievement, and have received influenza vaccination. Flutracking has a disproportionately lower number of participants who are Aboriginal or Torres Strait Islander [20].

It is important to note that the ILI rates captured by Flutracking are necessarily different from ILI rates captured by, e.g., general practice sentinel surveillance systems [37], since the Flutracking rates represent symptomatic cases in the community. Flutracking operates for 26 weeks each year and the completion rate is very high [20], so an ILI rate of 1 case per 50 completed surveys means that one ILI event is reported (at some stage during the season) for about every two survey participants. Note, however, that some participants may report multiple ILI events over the course of a single season.

### 4.2. Estimating the Probability of an Influenza Test, Given ILI Symptoms

We used a Beta distribution to estimate the probability that a Flutracking participant with ILI symptoms sought healthcare and had a specimen collected for testing, because it is a suitable model for the distribution of a probability value. If we treat each participant with ILI symptoms as an independent binomial trial, where success is defined as that participant reporting they had a specimen collected for testing, then we obtain a Beta posterior distribution from a Beta prior distribution by conditioning on the results of each weekly Flutracking survey.

We specified separate prior means ($\mu_{0}$) for each state, based on the 2014 Flutracking data for that state, and used a common prior variance ($\sigma^{2}$), in order to determine the initial shape parameters ($\alpha_{0}$ and $\beta_{0}$):(1)$$\begin{array}{cc} \sigma^{2} & =10^{-4} \end{array}$$
(2)$$\begin{array}{cc} \alpha_{0} & =\mu_{0}\cdot \left[\frac{\mu_{0}\cdot \left(1-\mu_{0}\right)}{\sigma^{2}}-1\right] \end{array}$$
(3)$$\begin{array}{cc} \beta_{0} & =\left(1-\mu_{0}\right)\cdot \left[\frac{\mu_{0}\cdot \left(1-\mu_{0}\right)}{\sigma^{2}}-1\right] \end{array}$$
(4)$$\begin{array}{cc} P(\mathrm{Test}|\mathrm{ILI},\mathrm{week}=0) & ∼\mathrm{Beta}(\alpha_{0},\beta_{0}) \end{array}$$
(5)$$\begin{array}{cc} E\left[\mathrm{Test}|\mathrm{ILI},\mathrm{week}=0\right] & =\mu_{0} \end{array}$$

With $N_{w}$ ILI cases and $T_{w}$ self-reported tests in the Flutracking survey data for week *w*, we obtain the Beta posterior distribution for the probability that a Flutracking participant with ILI symptoms would seek healthcare and have a specimen collected for testing:(6)$$\begin{array}{cc} P(\mathrm{Test}|\mathrm{ILI},\mathrm{week}=w) & ∼\mathrm{Beta}(\alpha_{w},\beta_{w}) \end{array}$$
(7)$$\begin{array}{cc} E\left[\mathrm{Test}|\mathrm{ILI},\mathrm{week}=w\right] & =\frac{\alpha_{w-1}+T_{w}}{\alpha_{w-1}+\beta_{w-1}+N_{w}}=\frac{\alpha_{w}}{\alpha_{w}+\beta_{w}} \end{array}$$
(8)$$\begin{array}{cc} \alpha_{w} & =\alpha_{w-1}+T_{w} \end{array}$$
(9)$$\begin{array}{cc} \beta_{w} & =\beta_{w-1}+N_{w}-T_{w} \end{array}$$

Note that as the total number of reported ILI cases increases, the expected value of the posterior approaches the observed proportion of ILI cases who reported having a specimen collected:(10)$$\lim_{w\to \infty }E\left[\mathrm{Test}|\mathrm{ILI}\right]=\lim_{w\to \infty }\frac{\alpha_{0}+\sum_{i=1}^{w}T_{w}}{\alpha_{0}+\beta_{0}+\sum_{i=i}^{w}N_{w}}=\frac{T}{N}$$

### 4.3. Influenza Surveillance Data

For three of the cities for which we generated forecasts—Brisbane, Melbourne, and Sydney—we obtained influenza case notifications data for the 2014–2017 calendar years from the relevant data custodian. These data include only those cases which meet the Communicable Diseases Network Australia (CDNA) case definition for laboratory-confirmed influenza [38], and represent only a small proportion of the actual cases in the community [39].

For Perth, we obtained PCR-confirmed diagnoses from PathWest Laboratory Medicine WA (the Western Australia public pathology laboratory) for the 2014–2017 calendar years. In comparison to the notifications data obtained for the other cities in this study, these data do not include serologic diagnoses. They also contain a much higher proportion of hospitalised cases (40–60%), which is further magnified because the bulk of PathWest community specimens come from regional and remote areas. People whose symptoms are sufficiently severe to result in hospitalisation will most likely seek healthcare, regardless of any other behavioural influences, and so these data are likely to exhibit little sensitivity to behaviour changes characterised in the Flutracking survey data.

In contrast to the other three cities in this study, we did not obtain all influenza case notifications reported to the relevant data custodian for the city of Perth. This was because we had only established a data-sharing agreement with PathWest Laboratory Medicine WA at the time of this study, rather than with the data custodian as per the other cities. It is clearly beneficial to use concordant data for each city, and we are working towards this for Perth and for other major Australian cities.

Many specimens do not include a symptom onset date, in which case we used the specimen collection date as a proxy. The difference between the time of symptom onset and the collection date is assumed to be less than one week for the majority of cases, and since we aggregate the case counts by week, we assume that this does not substantially impact the resulting time-series data.

Note that in this study we used the weekly case counts as reported after the 2017 influenza season had finished and all tests had been processed.

### 4.4. Forecasting Methods

We generated near-real-time forecasts by combining an SEIR compartment model of infection with influenza case counts through the use of a bootstrap particle filter [3,9,23].

First, we constructed a multivariate normal distribution for the peak timing and size in years prior to 2017, to characterise our prior expectation for the 2017 season. Parameter values for the SEIR model were then sampled from uniform distributions, and each sample was accepted in proportion to the simulated epidemic’s probability density. See Appendix A for details.

We modelled the relationship between disease incidence in the SEIR model and the influenza case counts using a negative binomial distribution with dispersion parameter *k*, since the data are non-negative integer counts and are over-dispersed when compared to a Poisson distribution [3].

The probability of being *observed* (i.e., of being reported as a notifiable case) was the product of two probabilities: that of becoming infectious ($p_{\inf}$), and that of being identified ($p_{\mathrm{id}}\left(w\right)$, the probability of being symptomatic, presenting to a doctor, and having a specimen collected). The probability of becoming infectious was defined as the fraction of the *model* population that became infectious (i.e., transitioned from *E* to *I*), and subsumed symptomatic and asymptomatic infections:(11)$$\begin{array}{cc} p_{\inf}\left(w\right) & =S(w-1)+E(w-1)-S(w)-E(w) \end{array}$$
(12)$$\begin{array}{cc} p_{\mathrm{ili}}\left(w\right) & =p_{\inf}\left(w\right)\cdot p_{\mathrm{id}}\left(w\right)+\left[1-p_{\inf}\left(w\right)\right]\cdot p_{\mathrm{bg}} \end{array}$$

Values for $p_{\mathrm{id}}\left(0\right)$ and *k* were informed by retrospective forecasts using surveillance data from previous seasons [3], while the background rate $p_{\mathrm{bg}}$ was estimated from out-of-season levels (March to May, 2017). The climatic modulation signals F(w) were characterised by smoothed absolute humidity data for each city in previous years, as previously described [24].

To calibrate the forecasts at each week *w*, we scaled the reference testing probability $p_{\mathrm{id}}\left(0\right)$ in proportion to the change in the expected testing probability (relative to the prior expectation), using a scaling coefficient κ. When the change was negative (i.e., the expected testing probability was less than the prior expectation) we instead raised the change to the power κ, to ensure that the calibrated value $p_{\mathrm{id}}\left(w\right)$ was strictly positive:(13)$$\begin{array}{cc} p_{\mathrm{id}}\left(w\right) & =p_{\mathrm{id}}\left(0\right)\times \left[1+f_{\kappa }\left(\frac{E\left[\mathrm{Test}|\mathrm{ILI},\mathrm{week}=w\right]-\mu_{0}}{\mu_{0}}\right)\right] \end{array}$$
(14)$$\begin{array}{cc} f_{\kappa }\left(x\right) & =\left\{\begin{array}{cc} \kappa \cdot x & \mathrm{if}\;x\geq 0\\ {\left(x+1\right)}^{\kappa }-1 & \mathrm{otherwise} \end{array}\right. \end{array}$$

As required, this is a smooth function of the change in the expected testing probability. We recovered the non-calibrated forecasts by setting κ=0, in which case $p_{\mathrm{id}}\left(w\right)=p_{\mathrm{id}}\left(0\right)$, as shown in Figure 3.

### 4.5. Forecast Performance

Forecast performance for each value of κ was measured via Bayes factors [24,25], relative to the non-calibrated forecasts. This is defined as the ratio ${\tilde{B}}_{k}\left(w\right)$, for each week *w*, of: (a) the likelihood of the future case counts $y_{w+1:N}$ according to the calibrated forecasts (κ=k); and (b) the likelihood of the future case counts according to the non-calibrated forecasts (κ=0):
(15)$${\tilde{B}}_{k}\left(w\right)=\frac{P(y_{w+1:N}|y_{0:w},\kappa =k)}{P(y_{w+1:N}|y_{0:w},\kappa =0)}$$

More generally, we can measure forecast performance relative to any value *K* for κ:(16)$${\tilde{B}}_{k,K}\left(w\right)=\frac{P(y_{w+1:N}|y_{0:w},\kappa =k)}{P(y_{w+1:N}|y_{0:w},\kappa =K)}$$

Our interest is in the average performance over the set of all forecasting weeks *W*:(17)$$\begin{array}{cc} {\tilde{B}}_{k} & =\frac{1}{\left|W\right|}\sum_{w\in W}{\tilde{B}}_{k}\left(w\right) \end{array}$$
(18)$$\begin{array}{cc} {\tilde{B}}_{k,K} & =\frac{1}{\left|W\right|}\sum_{w\in W}{\tilde{B}}_{k,K}\left(w\right) \end{array}$$

Note that the forecasting weeks *W* are only a subset of those weeks for which we have Flutracking survey data and influenza surveillance data.

Values in excess of 10 are considered as positive or strong evidence that the chosen model is more strongly supported by the data, and values in excess of 100 are considered as very strong (“decisive”) evidence [25,40]. Likewise, values less than −10 and less than −100 indicate strong and very strong evidence that the non-calibrated forecasts are more strongly supported by the data.

Here, we consider the logarithms to base 10 of ${\tilde{B}}_{k}$ and ${\tilde{B}}_{k,K}$, which we, respectively, denote ${\tilde{b}}_{k}$ and ${\tilde{b}}_{k,K}$. The evidence thresholds for ${\tilde{b}}_{k}$ (and also ${\tilde{b}}_{k,K}$) are therefore:(19)$$\begin{array}{cc} |{\tilde{b}}_{k}|\geq 1 & \Rightarrow \mathrm{strong}\;\mathrm{evidence} \end{array}$$
(20)$$\begin{array}{cc} |{\tilde{b}}_{k}|\geq 2 & \Rightarrow \mathrm{very}\;\mathrm{strong}\;\mathrm{evidence} \end{array}$$
where the sign indicates whether the data support the calibrated forecasts (positive values) or the non-calibrated forecasts (negative values).

To measure the *relative* performance of different values for κ, we also define the *normalised* logarithms ${\hat{b}}_{k}$ and ${\hat{b}}_{k,K}$, expressed as percentages of the maximum absolute value obtained over the chosen set of values *V* for κ:(21)$$\begin{array}{cccc} {\hat{b}}_{k} & =100\times \frac{{\tilde{b}}_{k}}{\left|{\tilde{b}}_{M}\right|}\; & M & ={\arg\max}_{v\in V}\left|{\tilde{b}}_{v}\right| \end{array}$$
(22)$$\begin{array}{cccc} {\hat{b}}_{k,K} & =100\times \frac{{\tilde{b}}_{k,K}}{\left|{\tilde{b}}_{N,K}\right|}\; & N & ={\arg\max}_{v\in V}\left|{\tilde{b}}_{v,K}\right| \end{array}$$

### 4.6. Availability of Materials and Methods

The data and source code used to perform all of the forecast simulations and generate the results presented in this manuscript are published online at https://dx.doi.org/10.26188/5bf3d60c4ffae, under the terms of permissive licenses. Please read the included file README.md for step-by-step instructions.

## Figures and Tables

**Figure 1 tropicalmed-04-00012-f001:**
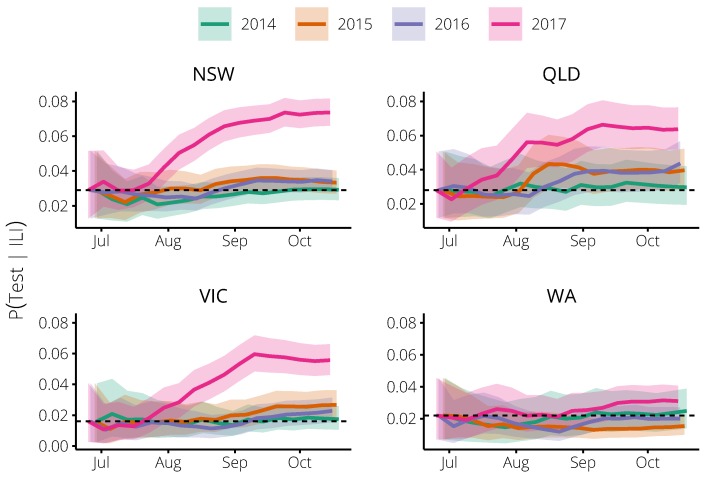
The probability of being swabbed for an influenza test, given ILI symptoms, as estimated from the weekly Flutracking survey data in each of the 2014–2017 seasons for participants in: New South Wales (**top-left**); Queensland (**top-right**); Victoria (**bottom-left**); and Western Australia (**bottom-right**). The horizontal dashed lines indicate the prior expectation for each state, the solid lines indicate the posterior expectation, after applying our statistical model, and the shaded regions indicate the 95% median-centred credible interval. In all states except Western Australia, this probability was much higher in 2017 than in previous years. Queensland also exhibited a moderate increase in 2015.

**Figure 2 tropicalmed-04-00012-f002:**
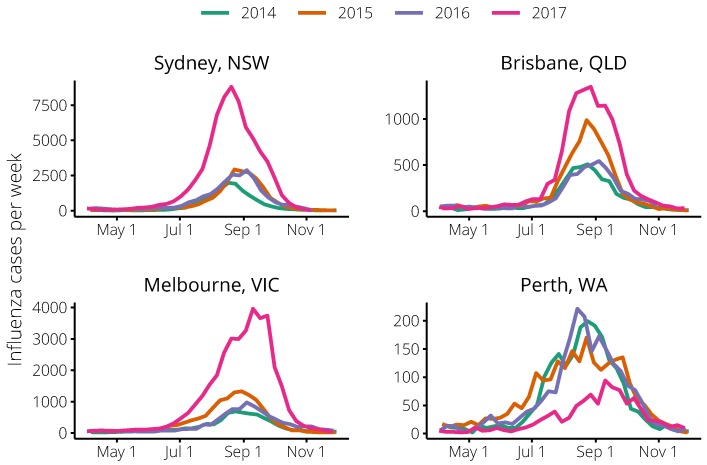
Weekly laboratory-confirmed influenza cases reported in each of the 2014–2017 seasons for the cities of: Sydney, New South Wales (**top-left**); Brisbane, Queensland (**top-right**); Melbourne, Victoria (**bottom-left**); and Perth, Western Australia (**bottom-right**). In all states except Western Australia (where the data comprise a greater proportion of hospitalised cases, see main text), the 2017 season was much larger than the 2014–2016 seasons. Queensland also experienced quite a large season in 2015.

**Figure 3 tropicalmed-04-00012-f003:**
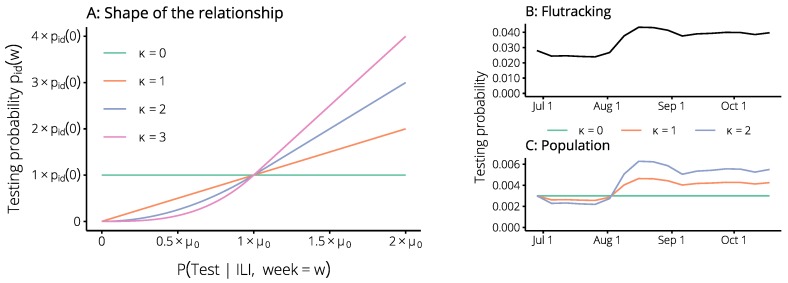
The testing probability $p_{\mathrm{id}}\left(w\right)$ for the general population is a smooth function of the testing probability $P\left(\mathrm{Test}|\mathrm{ILI},\mathrm{week}=w\right)$ obtained from the Flutracking data (panel A). Note that this relationship is non-linear as the Flutracking testing probability approaches zero, in order to ensure that the population testing probability remains non-negative, and that the Flutracking testing probability is defined in terms of its initial value $\mu_{0}=P\left(\mathrm{Test}|\mathrm{ILI},\mathrm{week}=0\right)$. The testing probability we obtain from the Flutracking survey data changes from one week to the next (panel B shows the values for Brisbane in 2015). The population testing probability will have a similar shape, with the scale controlled by the value of κ (panel C shows the values for Brisbane in 2015 for three values of κ).

**Figure 4 tropicalmed-04-00012-f004:**
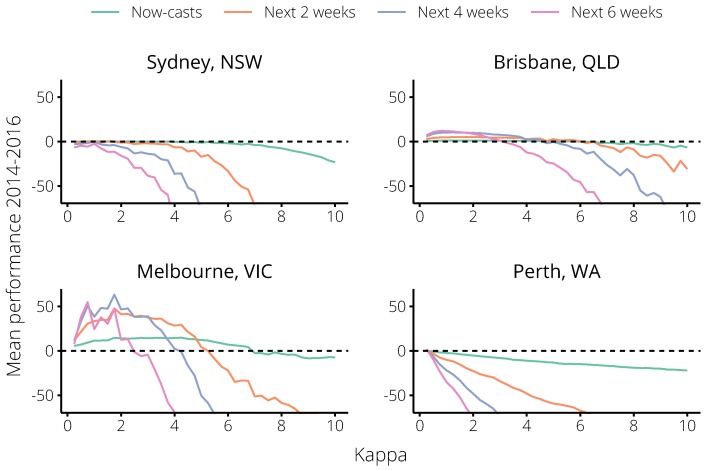
Mean forecast performance $\tilde{b_{k}}$ over the 2014–2016 influenza seasons, relative to the non-calibrated forecasts (the horizontal dashed line); $\tilde{b_{k}}>1$ indicates substantial performance improvement, and $\tilde{b_{k}}<-1$ indicates substantial performance reduction. Performance was measured against four different targets (see text for details), over a range of values for the scaling parameter κ. For small values of κ, forecast calibration significantly increased skill in Brisbane and Melbourne, had little effect in Sydney, and decreased skill in Perth.

**Figure 5 tropicalmed-04-00012-f005:**
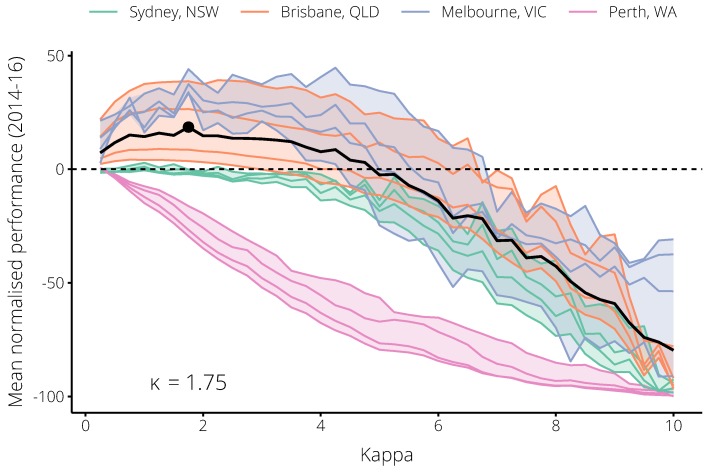
The trends in mean annually-normalised performance ${\hat{b}}_{k}$ for each city over the 2014–2016 influenza seasons, relative to the non-calibrated forecasts (the horizontal dashed line). Each set of coloured lines depicts the performance ${\hat{b}}_{k}$ for a city against the four chosen forecasting targets (see text for details). The solid black line is the overall trend for all cities *except* Perth; the optimal value of κ according to this trend is indicated by the black circle on this line (κ=1.75).

**Figure 6 tropicalmed-04-00012-f006:**
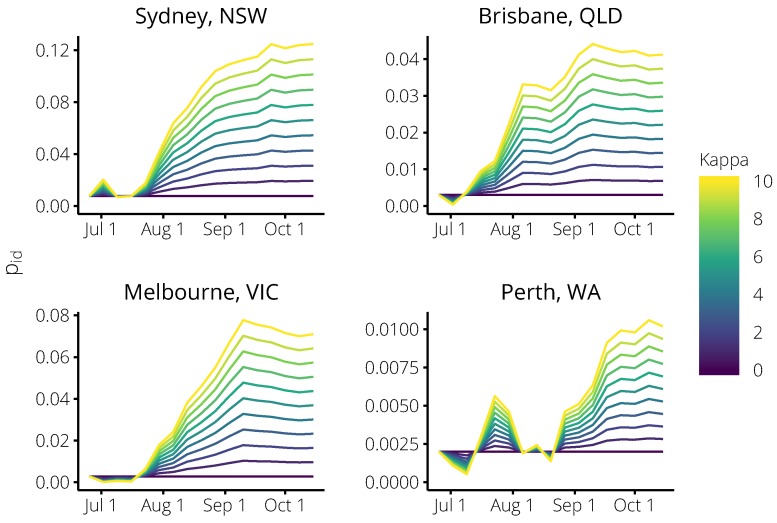
The trend in the testing probability $p_{\mathrm{id}}\left(w\right)$ at each week of the 2017 season, shown here for each city and over a range of values for κ. When κ=1 the (relative) change in $p_{\mathrm{id}}\left(w\right)$ is the same as the (relative) change in the Flutracking testing probability; when κ>1
$p_{\mathrm{id}}\left(w\right)$ is more sensitive to changes in the Flutracking testing probability; and when κ=0 we recover the *non-calibrated forecasts* (which do not account for any behaviour changes).

**Figure 7 tropicalmed-04-00012-f007:**
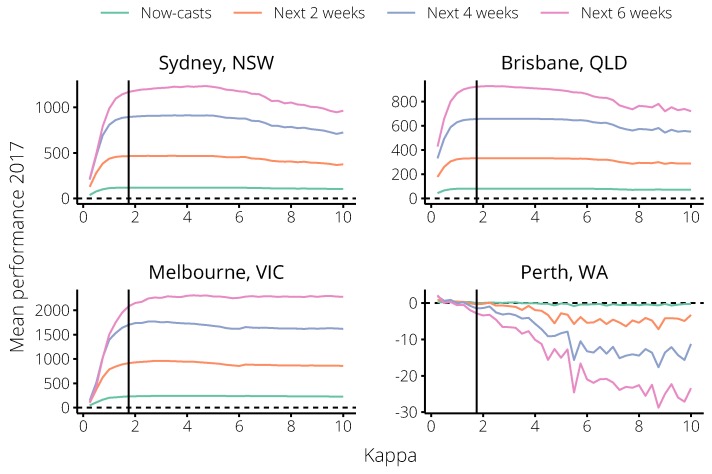
The trends in mean forecast performance ${\tilde{b}}_{k}$ over the 2017 influenza season for each city, relative to the non-calibrated forecasts (the horizontal dashed line); $\tilde{b_{k}}>1$ indicates substantial performance improvement, and $\tilde{b_{k}}<-1$ indicates substantial performance reduction. The black vertical line indicates κ=1.75, the value chosen as optimal based on the 2014–2016 seasons.

**Figure 8 tropicalmed-04-00012-f008:**
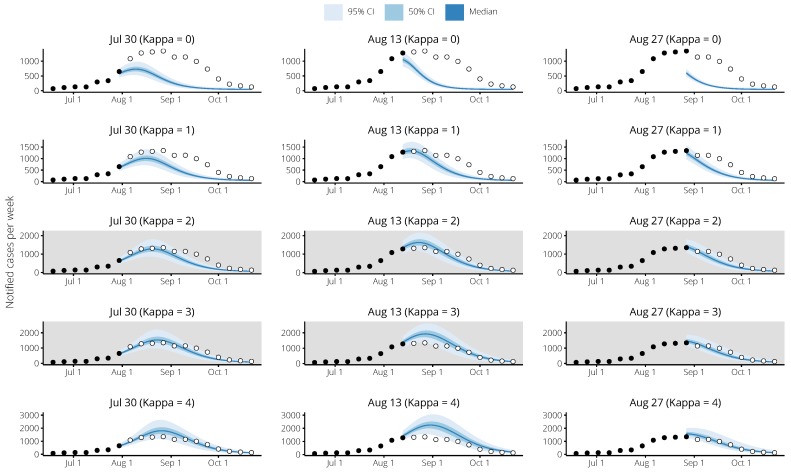
Example 2017 forecasts for Brisbane, Queensland, for various values of κ. Each column show forecasts generated for the same date, and each row shows forecasts generated using the same value of κ. Black points indicate reported case counts in weeks up to the time of the forecast, and hollow points indicate reported case counts in future weeks. The optimal values of κ for the Brisbane forecasting targets were 2–3 (shown with grey backgrounds), and this is supported by a visual inspection of the forecasts for each of the forecasting dates.

**Table 1 tropicalmed-04-00012-t001:** Flutracking mean weekly count of completed surveys, participants with ILI and participants with ILI who sought health advice and reported being tested for influenza (regardless of the test result), and the total number of influenza cases reported in each city; 2014–2017, NSW, Vic., Qld, and WA.

State	Year	Completed Surveys	Participants with ILI	Reported Tests	Influenza Cases
NSW	2014	6311	146	3	15,100
	2015	7242	155	4	23,324
	2016	8439	178	5	25,613
	2017	9405	218	14	71,752
Vic	2014	2815	64	1	7627
	2015	3348	68	2	13,566
	2016	3894	75	1	10,276
	2017	4712	104	5	37,739
Qld	2014	1736	36	1	5020
	2015	2180	46	2	8084
	2016	2481	50	2	5844
	2017	2822	68	4	13,365
WA	2014	1039	24	1	2302
	2015	3804	88	1	2515
	2016	3511	90	2	2399
	2017	3541	72	2	1087

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
