# Peer review of "Accounting for Healthcare-Seeking Behaviours and Testing Practices in Real-Time Influenza Forecasts"

_tropicalmed, 2019, doi:10.3390/tropicalmed4010012_

Round 1

Reviewer 1 Report

Excellent work on alternative systems of evaluation of epidemiological surveillance of influenza. The introduction is correct, the hypothesis and the objective are clearly explained.

The results, obtained through a retrospective analysis of an online community monitoring system (Flutracking) in four state capitals of Australia, are interesting. From a methodological point of view, the study is correct. The statistical analysis is coherent and adequate. The ethical aspects and limitations of the study are included.

In the conclusions, it is highlighted how human behaviour can act as a confusion factor in the interpretation of surveillance data and also the usefulness of obtaining key information through community surveillance systems to predict and evaluate epidemic trends in almost real time.

Tables and figures are correct.

Author Response

Please find our response in the attached PDF document.

Reviewer 2 Report

Thank you for giving me the opportunity to review the article. The authors conducted a study to account for healthcare-seeking behaviours and testing practices in real-time influenza forecasts in Australia. I thought that the topic is interesting, and the manuscript is well written. However, several concerns exist. Comments have been listed below.

Major Comments:

Abstract:

1.      L15: The authors should add the sentence(s) about the generalizability of this study results and the possibility of the practical usage in Australia and other countries.

Results:

2.      Figure 2: The authors reported influenza cases per week in the Figure 2, but the population adjusted number or infection rate may also be informative for the potential readers. If the information presented, readers can compare the shape of the Figure 1 and 2.

Materials and Methods:

3.      L230: The authors provided externally hosted supplementary files as an Appendix. However, files are not organized well, and it may be very difficult to reproduce the study results. So, the authors should provide a procedure manual of this study which clearly state the sequence of files (codes and datasets) used in the analysis.

Author Response

(The authors gave the same response as above.)

Reviewer 3 Report

The approach, methods, analyses and results are clear. I will focus my comments on the figures, which provide very important illustrations of results and interpretations of the analyses.

Figures 1 and 2 are straight forward and explained well.

Figure 3 is a very important figure and the right hand panel requires additional information to explain the data presented. The conclusions from this figure should be more clearly stated.

Figure 4 illustrates forecast improvement in Brisbane and Melbourne. This is interesting and some explanation of possible reasons for the differences would be interesting.

Figure 5 is described well.

Figure 6 is interesting. Even though testing probabilities did not increase in Perth, the forecast did improve using the correction factor. Why? Does this figure need to be included?

Figure 7 seems to contradict the previous figure although that may be that the interpretation of Figure 6 was not clear to me.

Figure 8 is essentially the capstone and demonstrates the forecast improvement. I believe the dotted line is actual cases reported - this needs to be stated in the caption statement.

The forecast methods and forecast performance calculations are described well.

Author Response

(The authors gave the same response as above.)

Round 2

Reviewer 2 Report

Thank you for giving me the opportunity to review the revised version of the article. I thought that the authors appropriately revised the manuscript according to the comments for the original manuscript.

Author Response

We are pleased to hear that reviewer 2 is happy that we have addressed their concerns.